# Exploring the feasibility of integrating health, nutrition and stimulation interventions for children under three years in Nepal's health system: A qualitative study

**Sophiya Dulal** [1]*, **Naomi M. Saville**[2], **Dafna Merom**[1], **Kalpana Giri**[3], **Audrey Prost**[2]

**1** Western Sydney University, School of Health Sciences, Sydney, Australia, **2** UCL Institute for Global Health, London, United Kingdom, **3** Health Research and Development Forum, Kathmandu, Nepal

\* dulal.sophiya@gmail.com

⊙ OPEN ACCESS

**Data Availability Statement:** Data supporting the findings are available in the article's supplementary material (S2 Table). The fully anonymised data can

## Abstract

Community-based primary care settings are a potential entry point for delivering Early Childhood Development (ECD) interventions in Nepal. Past studies have suggested that integrating stimulation with nutrition interventions is an effective way to deliver multiple benefits for children, but there is limited knowledge of how to do this in Nepal. We conducted a qualitative study in Nepal's Dhanusha district to explore how stimulation interventions for early learning could be integrated into existing health and nutrition programmes within the public health system. Between March and April 2021, we completed semi-structured interviews with caregivers (n = 18), health service providers (n = 4), district (n = 1) and national stakeholders (n = 4), as well as policymakers (n = 3). We also carried out focus group discussions with Female Community Health Volunteers (FCHVs) (n = 2) and health facility operation and management committee members (n = 2). We analysed data using the framework method. Respondents were positive about introducing stimulation interventions into maternal and child health and nutrition services. They thought that using health system structures would help in the implementation of integrated interventions. Respondents also highlighted that local governments play a lead role in decision-making but must be supported by provincial and national governments and external agencies. Key factors impeding the integration of stimulation into national programmes included a lack of intersectoral collaboration, poor health worker competency, increased workload for FCHVs, financial constraints, a lack of prioritisation of ECD and inadequate capacity in local governments. Key barriers influencing the uptake of intervention by community members included lack of knowledge about stimulation, caregivers' limited time, lack of paternal engagement, poverty, religious or caste discrimination, and social restrictions for newlywed women and young mothers. There is an urgent need for an effective coordination mechanism between ministries and within all three tiers of government to support the integration and implementation of scalable ECD interventions in rural Nepal.

**Funding:** This paper is a product of a PhD study supported by the School of Science and Health Postgraduate Research Scholarship and Western to the World Scholarship awarded to SD from Western Sydney University. The funders had no role in the study design, data collection and analysis, or decision to publish or prepare the manuscript.

**Competing interests:** The authors have declared that no competing interests exist.

## Introduction

In low- and middle-income countries (LMICs), 43% of children below five years of age are at risk of not reaching their full developmental potential due to co-occurring risk factors, including poverty, poor nutrition and inadequate opportunities for early learning [1–3]. Nurturing care, which includes good health, adequate nutrition, security and safety, responsive caregiving and opportunities for early learning, is a policy framework to holistically promote Early Childhood Development (ECD) [2,4]. Growing evidence suggests combining nutrition interventions with stimulation can promote ECD [5–7] but implementation strategies for doing this vary by context [8]. Stimulation refers to an interaction between young children and their caregivers that provides them with early opportunities to learn, play and communicate [9].

The World Health Organisation (WHO) recently developed guidelines to improve ECD, emphasising the importance of healthcare settings as a platform to scale up ECD interventions [10]. Research supports integrating stimulation interventions into health and nutrition services [4,11–14]. The health system encourages routine contact with caregivers and young children from pregnancy until early childhood, employs a diverse workforce, and provides health services to mothers and children at the health facility and community level [11]. This can be comprehensively leveraged to promote ECD more effectively. However, there is limited research on how to optimise the delivery of stimulation for early learning within routine health and nutrition services in LMICs [15].

Over the past two decades, the Government of Nepal (GoN) has made substantial investments in Early Childhood Education programmes for children aged three years and above through 35,991 early childhood education and development (ECED) centres [16]. Over two thirds of ECED centres are either in the community for children aged 3–4 years or in pre-primary classes within government schools for children aged 4–5 years. The remaining ECEDs operate within private schools [16,17]. To further expand ECD services, the GoN's Ministry of Education, Science and Technology (MoEST) developed a National ECD Programme (2004–2015), which aimed to implement multi-sectoral integrated interventions across education, health, nutrition, water, sanitation and hygiene and protection sectors to promote equitable ECD [16]. Despite these efforts, recent data from the Nepal Multiple Indicator Cluster Survey showed that the proportion of children aged 3–4 years who are developmentally on track for literacy and numeracy is low (40%), and only 56% of children are on track for socio-emotional development [18]. An evaluation of the National ECD Programme found that services were still provided in silos and rarely linked with an overarching ECD strategy [16]. Taken together, these data suggest there is inadequate coverage of nurturing care interventions for children below three years, highlighting the need for further work on ECD in Nepal.

Community-based primary health care settings are a potential entry point for delivering integrated ECD interventions in Nepal, but little is known about how best to do this or what enablers and facilitators might be. We aimed to understand the perspectives of community members, health service providers, stakeholders and policymakers on how best to implement integrated health, nutrition and stimulation interventions in rural Nepal. Specifically, the objectives of our study were to: a) understand caregivers' of children under three years, service providers', stakeholders' and policymakers' perceptions of integrated interventions; b) identify strategies to support the integration of stimulation interventions into existing health and nutrition programmes; c) understand the potential roles of governmental and non-governmental stakeholders; and d) identify barriers and facilitators to integration and community members' uptake of integrated interventions.

## Materials and methods

### Study setting

We conducted a qualitative study in Dhanusha district, lowland Nepal (*Terai*), from February-April 2021. Dhanusha district lies in Madhesh Province and comprises one sub-metropolitan city, eleven urban and six rural municipalities [19]. Most residents (89%) are Hindu, or Muslim (9%) Maithili-speakers [19]. Dhanusha has low female literacy levels at 40% [19]. Community-based health facilities include health posts and outreach clinics run by doctors, nurses, paramedics and Female Community Health Volunteers (FCHVs) [20], which provide large-scale health and nutrition services for children under five years [16,21].

### Sampling

We purposively selected one urban and one rural municipality with Maithili-speaking community members and sampled caregivers and health service providers. We also purposively sampled national and district stakeholders as well as policymakers to capture a variety of perspectives and triangulate findings. S1 Table provides sampling framework details.

From separate households, we sampled 18 caregivers (seven mothers, six fathers and five grandmothers) whose children/grandchildren were below three years for interviews. We purposively sampled caregivers from marginalised (Muslim and Dalit) and better-off caste groups and from poor and better-off households to explore a wide range of responses. Local FCHVs helped us to identify participants based on the inclusion criteria and introduced us to participants and their family members. Researcher SD did not know any FCHVs from the study site; however, another researcher KG was familiar with three FCHVs from the urban municipality. We did not disclose the topic for the interview with FCHVs prior to the participant identification.

We consulted with municipality leaders and persons in charge of health posts to identify 18 health service providers: two health coordinators, one health assistant, and one Auxiliary Nurse Midwife (ANM) were sampled for interviews and seven FCHVs from each municipality were sampled for FGDs. Likewise, we identified representatives involved in the implementation of health, nutrition, and education-related programmes including four national and one district stakeholders after consulting governmental and non-governmental organisation employees and sampled them for interviews. We sampled Health Facility Operation and Management Committees (HFOMC) members, six from each municipality for FGDs. We also sampled three policymakers from the health and education sectors of the national government and senior provincial health service representatives for interviews.

SD and KG approached community members in their homes and health service providers and district stakeholders at their workplaces. We communicated with national stakeholders and policymakers via phone calls and emails. We took voluntary informed written consent from all participants. Two caregivers (a father and mother) refused to take part in the interview because they felt shy to communicate with us. SD and KG assessed data saturation by reviewing interview recordings, field notes and daily debrief meetings and stopped recruitment once similar themes emerged from newly collected data [22].

### Data collection

We developed Semi-structured Interviews (SSIs) and Focus Group Discussions (FGDs) guide (S1 File) drawing on the WHO health system framework [23] and Nonadopting, Abandonment, Scale-up, Spread, and Sustainability framework [24]. However, we did not use either of

these frameworks to structure our results because we wanted to synthesise the data to inform policy development. All topic guides were developed in Nepali and Maithili.

We prepared a vignette of stimulation through play to explore participants' understanding of the concept of stimulation (S2 File) and piloted this with two mothers and one grand-mother. They were asked about the types of games played with their children and what children learned from them. We continued the process until participants made links between play and learning. We coined and used the phrase "play and learning" (*Khelne and sikne* in Nepali and *Khel aur sikai* in Maithili) when asking questions about stimulation interventions to caregivers and FCHVs. Following the vignette-inspired discussion, we moved on to questions in the topic guide.

Data were collected locally at a time and place chosen by participants. All caregivers were offered two handwashing soaps (~$0.5 USD), and their children were given a locally available toy (~$0.6 USD) in recognition of their participation. All interviews with stakeholders were conducted online via Zoom, except for three, which were conducted in person at participants' workplaces.

KG conducted all SSIs with caregivers (n = 17) and health service providers (n = 3) and FGDs with FCHVs (n = 2) and HFOMC members (n = 2) in the local language (Maithili), except for two SSIs (with a father and health coordinator) which were conducted in Nepali. SD conducted all SSIs with national (n = 3) and district (n = 1) stakeholders and policymakers (n = 3) in Nepali, except for one (with national stakeholder) in English. The language was based on the participant's preference.

SD took notes during each SSI and FGD to document non-verbal behaviour, descriptive information about the environment, and initial analytical points. SD observed and supervised KG during data collection and provided feedback after each SSI or FGD. SD did not suggest probes or follow-up questions during interviews as KG was well trained in formulating these. NMS visited the field for supervision, observed three interviews and provided feedback. During fieldwork, SD held briefs with KG daily and with AP and NMS weekly (online) to discuss emergent themes and changes required to the topic guide.

## Data management and analysis

All SSIs and FGDs were digitally recorded. SSIs averaged 75 (47–117) minutes, and FGDs averaged 133 (127–140) minutes in length. Nine SSIs and four FGDs conducted in Maithili were transcribed verbatim into Nepali by KG and translated into English for analysis by a fluent translator. All English-translated transcripts were proofread against the Nepali transcripts, with corrections made by SD for quality assurance. In addition, another local Maithili speaker directly transcribed and translated remaining 10 interviews conducted in Maithili into English and a second translator fluent in Maithili double-checked these for accuracy. Likewise, 10 interviews conducted in Nepali were transcribed and translated directly into English by SD. SD verbatim transcribed one interview in English.

We used the Framework approach described by Gale et al. 2013 to analyse the data [25]. We chose this method because it is flexible, promotes collaboration between analysts, and gives an audit trail of the analysis [25,26]. AP and SD read transcripts and made notes independently before discussing and agreeing on preliminary codes. For initial coding, we selected three thematically representative transcripts. AP, DM and SD coded these separately and met to discuss and agree on preliminary codes. We deductively and inductively developed themes and developed an analytic framework. SD then used this framework to index all the transcripts in NVIVO v20 software. Finally, SD charted the data on an excel spreadsheet to facilitate data mapping and interpretation. During data analysis, authors had regular meetings to explore

participants' responses further, discuss deviant cases, and agree on key themes. We revised and re-categorised themes and sub-themes until all authors agreed. We used the consolidated criteria for reporting qualitative research (COREQ) checklist to report the study procedures and findings (S3 File) [27].

### Reflexivity

This research forms part of SD's PhD. SD is a Nepalese female researcher with a nursing background, fluent in Nepali and English and with over ten years of research experience, including three years living in Dhanusha. SD has a basic understanding of spoken Maithili but required transcripts to be translated into Nepali and English for analysis. KG has completed higher secondary education and is a trained female qualitative researcher from Dhanusha. She is fluent in Nepali and Maithili, with over 15 years of research experience. SD and KG did not know or have any affiliation and relationship with the participants prior to the data collection. They maintained neutrality and kept their personal opinions and reactions aside during data collection and analysis.

### Ethical approval

We obtained ethical approval from the Western Sydney University Human Research Ethics Committee (H13813) and the Nepal Health Research Council (509/2020 PhD). We also sought approval from local authorities before starting participant recruitment, and informed written consent from all participants. We removed names and other personal identifiers from transcripts before analysis.

## Results

### Sample characteristics

Tables 1 and 2 describe the sociodemographic characteristics of caregivers/children and health service providers, district/national-level stakeholders and policymakers, respectively. We interviewed 18 caregivers (seven mothers, six fathers and five grandmothers), four health service providers, one district-level and four national-level stakeholders and three policymakers, (N = 30). We conducted two FGDs, each with FCHVs (N = 14) and HFOMC members (N = 12).

We organised our findings within three broad thematic categories, as summarised in Table 3. We present respondents' views in the text or in S2 Table. In Fig 1, we further summarise barriers and facilitators for the integration of stimulation interventions into existing national programmes and the use of the services by the community member using an adapted version of Bronfenbrenner's socio-ecological model [28].

Visual presentation of the barriers and facilitators for the integration of stimulation interventions into existing health and nutrition programmes within the health system of Nepal and the use of services by community members layered in the three levels of influence: caregiver-, health system-, and socio-cultural influences. ECD: Early Childhood Development; FCHVs: Female Community Health Volunteers; INGOs: International Non-governmental Organisations; NGOs: Non-governmental Organisations.

### The case for integrated interventions

**Many see nutrition and play are interrelated.**   Caregivers were interested in learning about the development of their children, learning how to play with them and how to incorporate this information into their daily life. While caregivers said that children need both

**Table 1. Socio-demographic characteristics of caregivers and children (N = 18).**

| Caregivers and children characteristics | n |
|---|---:|
| Caregiver relationship to child | |
| Mother | 7 |
| Father | 6 |
| Grandmother | 5 |
| Caregiver age (years), mean (SD) | 34.8 (13.3) |
| < 25 | 4 |
| 25–34 | 6 |
| 35–44 | 3 |
| > 44 | 5 |
| Caregiver education | |
| Never went to school | 6 |
| Primary | 3 |
| Secondary and above | 9 |
| Caregiver occupation | |
| Housewife | 5 |
| Farmer | 6 |
| Informal business | 5 |
| Labourer | 1 |
| Office clerk | 1 |
| Ethnicity/Caste | |
| Dalit/Muslim (most disadvantaged group) | 8 |
| Janajati/other terai caste (Sudi/Teli) | 8 |
| Yadav/Brahmin (least disadvantaged group) | 2 |
| Religion | |
| Hindu | 15 |
| Muslim | 3 |
| Age of child/grandchild (months) | |
| <12 months | 6 |
| 12–23 months | 8 |
| 24–35 months | 4 |
| Child sex | |
| Male | 10 |
| Female | 8 |
| | mean (SD) |
| Average number of children in the household | 4.2 (1.4) |
| Average number of adults in the household | 8.5 (4.5) |
| Average walking time from household to health facility (minutes) | 14.9 (9.9) |

Abbreviation: SD, Standard Deviation.

nutritious food and adequate play, several health workers, stakeholders and policymakers highlighted that nutrition and stimulation were interrelated and noted that combined interventions would make children mentally active, physically strong and healthy, and improve their future academic performance.

> *"Both nutrition and play programmes are like fingernails and muscles. It will be good to run both programmes combined. When children are given nutrition, they will just eat and sit. If*

**Table 2. Sample characteristics of health service providers, stakeholders and policymakers (N = 38).**

| Characteristics | Health service providers (N = 18) | n | District stakeholders (N = 13) | n | National stakeholders (N = 4) | n | Policymakers (N = 3) | n |
|---|---|---|---|---|---|---|---|---|
| **Roles** | Health coordinator | 2 | Nutrition programme manager | 1 | ECD expert in civil society organisation | 1 | Province health representative | 1 |
| | Health assistant | 1 | Teacher | 2 | Senior technical advisor for nutrition and health at INGO | 1 | Senior public health officer | 1 |
| | Auxiliary nurse midwife | 1 | Ward administrative assistant | 2 | ECD specialist at UNICEF | 1 | Education officer | 1 |
| | Female community health volunteers | 14 | Health post in charge | 2 | Chief of the nutrition section at UNICEF | 1 | | |
| | | | Female member | 2 | | | | |
| | | | Male member | 2 | | | | |
| | | | Ward chairperson | 2 | | | | |
| **Age (in years), mean (SD)** | 47 (12.2) | | 45.1 (12) | | 49 (13.6) | | 46 (10.5) | |
| **Gender** | | | | | | | | |
| Male | 3 | | 11 | | 3 | | 2 | |
| Female | 15 | | 2 | | 1 | | 1 | |
| **Highest education level** | | | | | | | | |
| None | 5 | | 2 | | | | | |
| Primary | 2 | | | | | | | |
| Secondary | 8 | | 3 | | | | | |
| Higher secondary and higher | 3 | | 8 | | 4 | | 3 | |
| **Average number of years employed in role, mean (SD)** | 19.1 (3.5) | | 4.6 (8.1) | | 7.3 (1.9) | | 3.5 (3.0) | |

Abbreviations: ECD, Early Childhood Development; HFOMC, Health Facility Operation and Management Committee; INGO, International Non-Governmental Organisation; SD, Standard Deviation; UNICEF, United Nations Children's Fund.

*they are only taught to play, they will not get enough nutrition. If they are taught to play, then when they sweat, they will need nutritious food to provide strength to their body. Children will get tired when they spend energy and become weak and need nutritious food."* Health assistant, SSI 22

**Informal play already occurs in the community, but is not always connected to learning.** Mothers and grandmothers described existing activities with children including showing them how to play with toys, teaching them to speak, holding their hands to help them walk, oil-massaging their bodies, telling stories, and singing songs. Health workers, stakeholders and a policymaker noted that caregivers play with children but are unaware of any benefits to their children's development. Although fathers thought play was important for their children, they mostly purchased local toys but did not play with them.

Caregivers and FCHVs said mothers are busy with household chores, agricultural work or jobs so only play with their children during leisure time. Caregivers viewed play as an activity child do naturally for amusement and mostly did alone, with siblings or with other children. Some caregivers said playing kept children engaged and prevented them from bothering their parents. Parents were more focused on educating their children than on playing.

**Gaps in the implementation of the existing policy for integrated ECD interventions.** All caregivers, health service providers and district stakeholders said that there was no intervention to promote stimulation for children under three years old. Likewise, national

**Table 3. List of identified themes and sub-themes.**

| Themes | Sub-themes |
|---|---|
| The case for integrated interventions | Many see nutrition and play are interrelated |
| | Informal play already occurs in the community, but is not always connected to learning |
| | Gaps in the implementation of the existing policy for integrated ECD interventions |
| | Existing health and nutrition programmes provide opportunities for integration |
| Integrated interventions in practice | Intervention orientations and advertisements to promote participation |
| | Focus on mothers, with the involvement of other family members |
| | Participants' expectations of incentives should be addressed |
| | Involving health workers and FCHVs in delivery |
| | Provide interventions locally with a dedicated space to improve access |
| | Health posts or outreach clinics in communities are good locales for delivery |
| | Strengthening monitoring and evaluation strategies |
| | Mobilise local opinion leaders to overcome religious and caste discrimination |
| | Engage household heads and officials to encourage the participation of young women |
| | Support poor households to ensure their participation |
| Support from national and local governments | Prioritise funding for ECD at the national level |
| | Recruiting new health workers |
| | Review training, supervision and incentives for health service providers |
| | Local governments play a key role |
| | Governments can get support from external agencies |

Abbreviations: ECD, Early Childhood Development; FCHVs, Female Community Health Volunteers.

stakeholders and policymakers reported that community ECED centres focus on children over three and do not incorporate play in promoting early learning.

National stakeholders and policymakers said the national government had recently published a national ECD strategy to guide local governments to plan ECD interventions and support the holistic development of children through intersectoral coordination. However, they claimed that federal ministries did not consider stimulation interventions a priority, which has promoted a culture of siloed working at the ministry level.

> *"We need one integrated programme for ECD. This is what we are looking for. A new ECD strategy paper has also conceived this idea that we need such programme. This is not only the federal government's responsibility. This should be implemented by all three levels of government: federal, provincial and local. Within these governments, for example, in Kathmandu city, there are six types of ministries that are directly related but what they do is that they work separately . . . there is no linkage between the ministries. But now the strategy paper is raising these things."* National stakeholder, SSI 30

Several national stakeholders and a policymaker said that the Ministry of Health and Population (MoHP) had introduced stimulation into the maternal, infant and young child feeding (MIYCF) programme. While some districts/municipalities had begun implementation, training was delayed due to COVID-19.

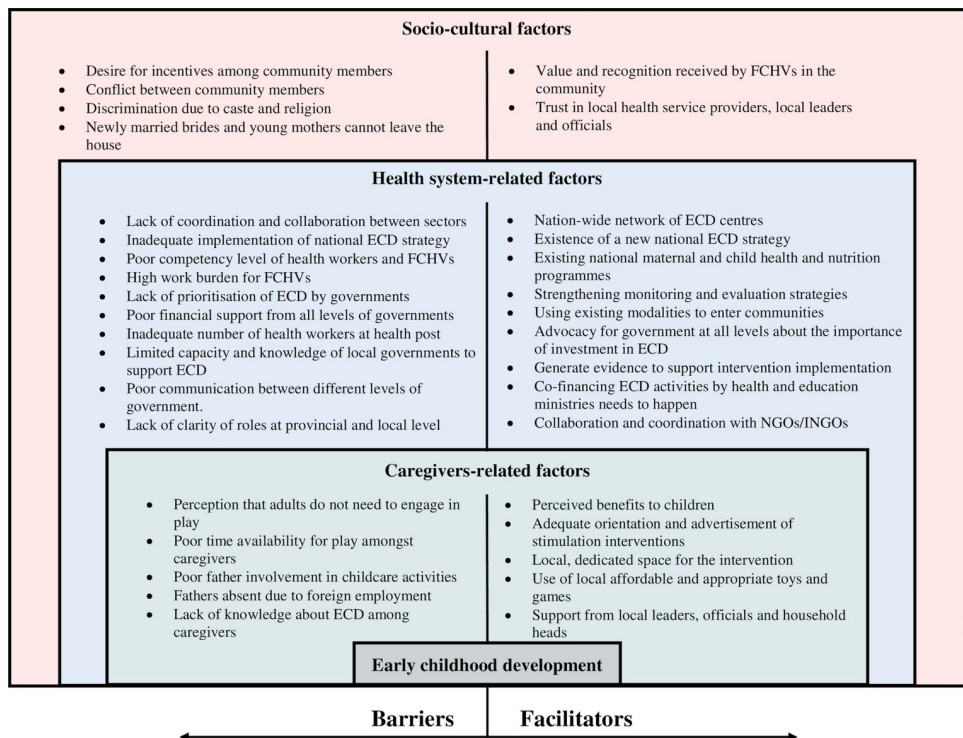

**Fig 1. Factors influencing integration of ECD interventions within Nepal's health system and their uptake by communities.**

**Existing health and nutrition programmes provide opportunities for integration.** Most health service providers, stakeholders and policymakers stressed that stimulation interventions should be integrated into various facility- and community-level maternal and child health and nutrition programmes rather than standalone. These include antenatal and postnatal care visits, immunisation programmes, community-based integrated management of childhood illness and newborn care, Golden thousand days (nutrition from conception to 2 years), MIYCF programme, Integrated Management of Acute Malnutrition, Nutrition Rehabilitation Homes, distribution of therapeutic foods, vitamin A, multivitamins or anti-parasitic medicines and growth monitoring programme. Several national stakeholders suggested that reviewing the Multi-sectoral Nutrition Plan (MSNP) II would provide an excellent opportunity to integrate stimulation intervention into the national nutrition programmes.

## Integrated interventions in practice

**Intervention orientations and advertisements to promote participation.** Respondents consistently advocated that to encourage community members' participation in integrated interventions, awareness of the significance of stimulation for early learning should be raised through orientations and advertisements at community gatherings, healthcare visits, schools, ECED centres and via mass media like social media, radio, and television.

*"In the ward, there should be a meeting, and then information about the programme should be discussed and then only all the children will be involved. There should be a meeting in every tole [area] . . . advertisement should also be done, and only it is possible to encourage participation."* ANM, SSI 19

## Focus on mothers, with the involvement of other family members

Caregivers, health workers and district stakeholders consistently advocated for educating mothers about play techniques since they are primarily responsible for childcare. Health service providers, stakeholders and policymakers suggested involving women from early pregnancy and the postpartum period to promote child development from the first days of a child's life. Fathers were considered breadwinners and their role was to pay school fees and provide for basic needs, including food and clothes. Fathers were mostly busy with local jobs or overseas employment, often away from home for long periods and unable to participate in childcare.

> *"A father needs to learn as well, but in most houses, the father of the children in Nepal mostly goes for overseas employment. So, there is no chance for fathers to attend such programmes for three years."* Father, SSI 3

Some caregivers and health service providers said grandmothers should be included as they often take care of children when mothers are busy with household chores and agricultural work. Others proposed teaching playing techniques to siblings.

**Participants' expectations of incentives should be addressed.** Some caregivers indicated they would not participate in an intervention that did not include incentives such as toys, food, clothing, or cash. Health service providers and HFOMC members echoed this.

> *"If I am getting any facility, then only I will go; otherwise, why would I waste my time? Everyone does not have leisure time to spend for free (for nothing in return) . . . When people are tempted then only they will go."* Mother, SSI 10

Several health workers, stakeholders and a policymaker suggested that incentives that promote behaviours are reasonable, but participation-boosting incentives are counterproductive and unsustainable. To avoid expensive incentives, stakeholders and a policymaker suggested introducing new interventions through local governments and existing contact points.

> *"If parents come to participate, they would not like to go empty-handed. . . So, it would be better to integrate the programme with the ongoing ones. They will come with the desire to receive a vaccination, Vitamin A, or Baal vita (micronutrient supplementation). This way, our health and nutrition programme will be successful, and the play programme will go smoothly."* HFOMC members, FGD 3

**Involving health workers and FCHVs in delivery.** Many agreed that integrated interventions should be delivered by FCHVs because they understand the context, are from the same neighbourhood, trusted by community members and know about child health issues. However, health service providers, stakeholders and policymakers also highlighted that FCHVs might lack competency to independently support stimulation activities and should be supervised by health workers.

Health service providers and stakeholders identified ageing of existing FCHVs as a barrier to their ability to deliver the intervention. Many FCHVs are reluctant to retire because of the recognition they receive in the community. They are willing to take on new responsibilities and also want additional allowances from the government. One health worker and a national stakeholder said that the retirement policy should be enforced, with retiring FCHVs serving as mentors.

*"FCHVs will never say that their workload has increased and that they will not work. They will always say, "We will do it. We have been doing this for a very long time." because they have understood their face value and reputation in the community."* National stakeholder, SSI 29

Health workers and stakeholders said it could be challenging for FCHVs to perform additional tasks because they are already overburdened. To overcome this challenge, several stakeholders suggested hiring new staff, such as community or ECD facilitators. However, other stakeholders thought recruiting additional staff could lead to conflict and was neither cost-effective nor sustainable. Meanwhile, some caregivers and health coordinators said health workers should deliver the intervention. One policymaker said the GoN intends to delegate current health services to ANMs, which will reduce the work burden of FCHVs.

**Provide interventions locally with a dedicated space to improve access.** Some caregivers, health workers and district stakeholders highlighted that local individuals should deliver the integrated interventions in the local language and that doing this near their village would encourage community members to participate.

Caregivers, health service providers and HFOMC members said that, to encourage participation, local government should provide facilities, including physical space, mats and tarpaulins to sit on during a gathering, chairs and benches at outreach clinics, a private room for breastfeeding mothers and restrooms. Similarly, most caregivers, health service providers and district stakeholders suggested identifying a dedicated space for play, such as a playground or a park, supplied with toys and food for children, and a teacher to lead play activities.

*"In my village, there should be a small place for children to play and any materials to play with should be provided because a child under three years of age starts to play and, at that age, they will learn ABCD, or they will play. They will learn through games. If that could happen, it would be good because a child under three years won't go to school to study."* Father, SSI 3

**Health posts or outreach clinics in communities are good locales for delivery.** Health coordinators, stakeholders and a policymaker said that existing health service modalities, including contact points at health posts or outreach clinics, provide opportunity to deliver integrated interventions:

*". . .when a parent brings a child to the health facilities or the health post for another service, that health worker can have, even if for just a two-minute, a conversation with caregivers about the role early stimulation can play in helping in the further development of a child and maybe childhood development milestones."* National stakeholder, SSI 27

Most respondents including caregivers preferred delivery via the government's existing mothers' groups because they felt comfortable speaking there, received peer support, and felt that groups enabled new information to spread easily and rapidly. Respondents did not favour home visits which were felt to be time-consuming, costly, and with too many distractions for families to focus, as explained by FCHVs:

*"I: You did not select the option of a home visit. Why? P1: If we do a home visit, we must give more time [. . .] P2: In the house, they have their work [. . .] they will think of their work and will not listen to us. P6: If I visit their house, they will say they have to do their work and will not give time. P1: . . . How many houses can we go to if we visit each one? It will take more time."* FCHVs, FGD 1

However, some felt that home visits were appropriate for mothers with children below six months, those uncomfortable sharing information in a group, and for marginalised groups that would feel excluded from groups by higher caste members.

Some caregivers, a health worker and stakeholders said that audio-visual such as short films and pictures were easy to understand for mothers with low literacy. Others suggested demonstrating play techniques and having mothers/carers practice with their children would be the best way to communicate. Respondents emphasised the importance of local toys and games:

*"Local games should be included, not the foreign-based games, which parents cannot afford to manage tomorrow. What can be done in the local context that should be taught, then they will be able to practice at home. They will be able to manage toys. We need to capacitate in that manner."* District stakeholder, SSI 23

**Strengthening monitoring and evaluation strategies.**   Health service providers, stakeholders and policymakers reported that the government should strengthen monitoring and evaluation mechanisms by expanding the skilled workforce and ensuring transparency and accountability. Some said that health workers need to be upskilled through orientation sessions to update their knowledge and skills on monitoring tools and techniques. Others suggested that stimulation indicators should be added to existing national and state-level surveys to improve transparency and accountability in intervention delivery. A district stakeholder suggested developing guidelines for a follow-up system to monitor actual practices in the community.

**Mobilise local opinion leaders to overcome religious and caste discrimination.**   Some caregivers, FCHVs, stakeholders and policymakers said that notions of untouchability could exclude marginalised groups. Some caregivers noted that different castes would not mix and may feel conflicted about having a separate intervention for their caste.

*"For children, people from different castes may fight, saying, "I will open a different group for my child, or you should open a different group for your child." That might happen. They may fight, saying, "I will not let my children participate in this." Like I am a "Sahu" [landlord], and I will not allow "Doom" [Dalits] children to play with mine. I will not let my children play with Muslim kids."* Father, SSI 6

To overcome this barrier, health workers and stakeholders suggested that formal and informal leaders, including teachers, imams, priests, and traditional healers, be taught about the intervention first, and the community then be approached through them.

**Engage household heads and officials to encourage the participation of young women.**   Some caregivers, health service providers and district stakeholders said that newlywed brides and new mothers were not permitted to leave the house and would not be able to participate in the intervention. They suggested that household heads (often mothers-in-law), as the primary decision-makers, should be informed about the initiative to increase new mothers' participation. Most caregivers, health service providers and HFOMC members agreed that having a health coordinator, mayor from the municipality, or ward representative oversee the intervention, especially in the beginning, would help increase community members' trust and through this boost the participation of young women. Moreover, community members are influenced by local and well-respected people, including teachers, health workers, social workers, and political representatives.

**Support poor households to ensure their participation.**   Some caregivers, FCHVs and stakeholders said poverty impedes caregivers' ability to provide food, toys, and education for

their children, and argued that the government must support them. As parents need to work to provide for their families, they necessarily give less priority to spending time with their children and will not be able to participate.

*"[. . .] if I am unemployed and have no job, my house will run only after I earn some money. So, earning is necessary. For instance, I am educating my child, although I don't have that status. Still, I want my child to get educated. But if I won't have that much money, how will I educate them and so I must work. . . If I don't work, things will not work for me. . . So, I will not be able to manage my time like whether go to job or those programmes."* Father, SSI 1

## Support from national and local governments

**Prioritise funding for ECD at the national level.**   Some health service providers, stakeholders and policymakers highlighted that governments do not prioritise investment in ECD as other sectors take precedence over those most directly relevant to nurturing care:

*". . . according to the people's representatives (the focal person in any development activities in the community), development does not mean the health sector. Health is not related to development. According to them, infrastructure, roads and irrigation are considered development factors. For instance, we had a massive cut in the budget allocated for the health sector only. [. . .] They do not consider child development an important subject. . ."* Health coordinator, SSI 20

Several stakeholders and a policymaker mentioned that the budget allocation should be based on the intensity of the problem and on evidence of what works. Other national stakeholders said deferred rollout of the government's budget and procurement system delayed intervention implementation. Ministries could jointly allocate funds for integrated interventions at the national level.

*". . . if we want to take the ECD and nutrition activities at the community level, then we can also manage joint financing. That means we need to ask the Ministry of Education how much they can invest because they can also invest from the national level. Similarly, how much investment can the Ministry of Health make because we might also take some components through FCHVs."* National stakeholder, SSI 29

**Recruiting new health workers.**   A health coordinator, a national stakeholder and a policymaker were concerned about increasing the workload of cadres in understaffed health facilities. Some municipalities recruit locally, but these recruits lack knowledge and are not skilled. Several health workers, stakeholders and a policymaker suggested increasing health workers' qualification criteria and clarifying their roles and responsibilities to enhance coordination.

*"The current health workers in the health post were appointed based on the workload from the previous 30 years. In the current situation, we cannot function unless we hire new health workers. It seems like we need twice as many as we currently have. For this, provincial and local governments have hired temporary workers to help with this. . . They have been hiring 4–5 staff recently."* Policymaker, SSI 25

**Review training, supervision and incentives for health service providers.**   Several health service providers, stakeholders and policymakers suggested giving refresher and in-service training to health workers and FCHVs to improve knowledge retention, motivation, and

service quality. A health coordinator, a national stakeholder and a policymaker said high turn-over of master trainers results in repeated training and increased costs. Further, training courses are crowded and short, hindering participants' ability to understand and retain information.

> "... we provide training to FCHVs for two days to orient them. They learn and go to the field. They can get confused in the field because more than 20–25 people had participated in that training. Everyone has a different rate of understanding in those two days of training." Policy-maker, SSI 25

According to some health service providers, stakeholders, and policymakers, monetary incentives and performance-based rewards could motivate health service providers to be more accountable. On the other hand, a policymaker mentioned that differences in financial incentives offered by government and external agencies had affected the sustainability of interventions. To overcome this issue, external partners are encouraged to follow government norms.

**Local governments play a key role.**   Stakeholders, health workers and policymakers said local governments should plan, implement and monitor community-level integrated ECD interventions. Implementing any intervention at the community level is challenging as new local governments adjust to decentralised federal governance. A health coordinator said many municipality staff are newly appointed and unclear about their role and responsibilities.

Some stakeholders and policymakers highlighted a lack of communication and coordination at national, provincial and local levels. They said that the provincial governments lack clarity in their role and cannot support local governments, which then falls to national stakeholders.

> "Local governments can plan and do what they want, but they do not have the technical expertise about what to do. They are looking for the federal government for their guidance ... There is a need for a lot of capacity development of the local government to contextualise and adapt and turn the national guidelines and strategies into actions." National stakeholder, SSI 27

Despite this, stakeholders and policymakers noted that local governments were willing to learn and implement new interventions in their communities. Health service providers, stakeholders and policymakers suggested the national government should generate evidence on the effectiveness of integrated interventions within Nepal and develop policies incorporating stimulation into the national programme.

**Governments can get support from external agencies.**   Stakeholders and policymakers stressed the need for supporting agencies (NGOs and INGOs), to collaborate and coordinate closely with local governments to strengthen their capacity to deliver interventions, provide funds, human resources, materials, training, advocacy for ECD strategies and enhance multi-sector coordination. Several stakeholders and a policymaker said agencies could support the government in FCHV supervision.

> "Since we have few health officers, we (the government) cannot attend every mother group meeting. Where NGOs/INGOs staff are present if they could support those FCHVs running the group sessions or while interacting with the group in any new interventions. Provide guidance and supervision to FCHVs and help them. Sometimes if wrong messages are being delivered, then that gets corrected. This makes it easier and is seen as a good thing." Policymaker, SSI 25

Some caregivers, stakeholders and policymakers said NGOs and INGOs could provide technical support to gather evidence for integrated interventions because they have up-to-date knowledge. However, a policymaker said that any successful intervention would then need to be handed over to the public health system.

## Discussion

In this qualitative study, we found that respondents were positive about introducing stimulation for early learning into national maternal and child health and nutrition services and saw potential benefits for children's overall development. They recommended using existing contact points between community members and frontline workers and multiple delivery strategies to support integrated interventions. Local governments must play a key role in integrating stimulation into health and nutrition services, but support from provincial and national governments are essential. Health system barriers to integrating stimulation for early learning into Nepal's health system include a lack of intersectoral collaboration, poor competency among health workers, increased workload for FCHVs, insufficient financial support, lack of prioritisation of ECD by the government, and insufficient capacity within local governments. Caregiver-related and socio-cultural factors such as lack of knowledge and awareness, caregivers' limited time, gender roles, poverty, religion and caste discrimination, as well as social restriction on newlywed women and young mothers could limit community members' ability to participate in integrated interventions for ECD.

Respondents considered that stimulation activities could be integrated into many maternal and child health and nutrition programmes as well as in ECED centres. Healthcare visits that often involve long waiting times, for instance, immunisation or monthly growth monitoring programmes, are better suited to offer ECD activities [29,30] than rapid ones with limited opportunities for interaction with healthcare workers. While multiple sectors can provide feasible and acceptable entry points for ECD interventions [31], intersectoral coordination is necessary to facilitate smooth integration [32,33]. In Nepal, the MOHP and MoEST are responsible for health, nutrition and ECD activities but they work in silos [16], which significantly challenges intersectoral coordination [34]. While respondents shared that the new ECD strategy could bridge this gap and promote intersectoral coordination, previous similar efforts in Nepal have not succeeded. For instance, Nepal's MSNP aims to improve nutrition through an intersectoral approach. While the MSNP has strengthened the nutrition system and increased investments in nutrition in Nepal [35], there is lower nutrition expertise and weaker implementation at the district level [36] and little ownership in any sector but health, limiting collaboration within districts [37,38]. This means that intersectoral approach in Nepal would require strengthening administrative capabilities and political will at a provincial and local level.

Another important finding was that mothers are seen as the privileged interlocutor for all integrated ECD interventions. Patriarchal gender norms in Nepal and other settings frame childcare as a women's responsibility, entrenching fathers' roles as providers rather than carers [39], despite the fact that fathers' engagement in stimulation can also support mothers' caregiving practices [40] and influence child development [41,42]. Although there have been repeated calls to engage multiple family members in childcare [9,42–44], the primary focus of ECD interventions remains on the mother. In a recent systematic review of 24 studies of integrated interventions [7], only one study included fathers [45]. Interventions targeting only mothers to improve child development could potentially limit effectiveness and reproduce patriarchal models that confine women to domestic work and childcare. In Nepal, grandmothers are highly regarded in the family and play an essential role in decision-making for child health

and nutrition-related practices [46]. Involving grandmothers in ECD interventions could also help overcome traditional barriers such as female seclusion norms, where newlywed women and young mothers are not allowed to leave the house [47], and improve mothers' access to the intervention.

We found that CHWs, and particularly FCHVs, should play a key role in supporting integrated interventions for ECD. This finding echoes experiences from other settings. Studies from Bangladesh and Pakistan have shown that integrated interventions delivered by CHWs can have promising results [13,14]. In Kenya, CHWs were perceived to have a good knowledge of health, nutrition, and stimulation and as a reliable source of information [48]. Nepal's FCHVs are respected and trusted by community members [38]. They are deployed to address the shortage of health workers [49] and have contributed to improving community health outcomes [50,51]. However, FCHVs are assigned various health and sometimes non-health-related programmes [52] without due consideration for their existing capacity. Poor literacy, increasing workload, insufficient supervision and lack of monetary incentives [52,53], raise the question of whether deploying FCHVs to support integrated ECD interventions is feasible [51,54].

Our study highlights the opportunity to deliver stimulation activities via contact points in health posts and outreach clinics, as well as the use of group meetings. In Bangladesh and the Caribbean, stimulation interventions provided by local agents through groups and integrated with child health visits at health centres were effective [13,29,55] and cost-effective [29]. Advantages of group meetings mentioned in other studies [56–58] include mutual trust, friendship and peer support, which are consistent with our findings. However, marginalised groups favoured home visits to avoid the risk of exclusion. Nevertheless, home visiting in a dispersed rural setting can be labour-intensive and expensive to bring to scale [9]. The delivery strategy must consider the socio-cultural context and align with the existing services [59], so further research is necessary to identify the most feasible, acceptable and cost-effective strategies for Nepal's varying contexts. Our study provides a 'menu' of options to choose from and factors to consider at a provincial and local level.

Our study respondents suggested government support for poor households in accessing integrated interventions. Cash transfers linked to or offered alongside stimulation interventions have led to improvements in child's developmental outcomes [8,60,61]. A quasi-experimental study showed that a Child Grant parenting programme in 25 districts of Nepal improved overall developmental outcomes for children aged 0–5 years of age [62]. Further, by incorporating stimulation interventions into a cash transfer programme, families may be able to invest in nutrition, play materials, and educational tools for their children, contributing to ECD [12,63]. The implementation and evaluation of cash transfer programmes with integrated interventions could support ECD for children in resource-constrained settings like Nepal.

### Implications for practice and further research

Our study explored approaches to integrate stimulation for early learning into health and nutrition services in the rural plains of Nepal. Any initiative to improve ECD in this context should build on existing knowledge and routine childcare play practices, enhancing these and expanding their use to other family members, including fathers and grandmothers, not just mothers. Such initiatives can empower families, encourage programme ownership and facilitate programme success and sustainability [64]. Meanwhile, in parallel, interventions should provide spaces and times in the community or ECED centres where caregivers can comfortably bring children under three years to play. The availability of dedicated spaces for play in the community could make stimulation 'visible' and improve access. Future research should

include implementation and evaluative research on integrated ECD interventions in different contexts of Nepal to generate evidence on their effectiveness and cost-effectiveness.

Fathers' involvement in childcare can improve family members' participation and practice of parenting messages [56,65], encouraging fathers' involvement is essential to support mothers in childcare and promote ECD. To promote gender equity and family inclusion, it is important to educate fathers and grandparents about the benefits of stimulation, encourage them to participate in childcare through their abilities and skills, as well as support an equitable distribution of childcare and household chores. Improving health workers' knowledge and counselling skills to communicate accurate caregiving knowledge to fathers and grandparents could be helpful. Therefore, policy and programme development must focus on empowering both parents as well as family members to promote their children's holistic development.

Strengthening the health system requires capacity building of health service providers to improve the delivery and scalability of integrated interventions [58]. Management and availability of human resources have been challenging in Nepal's new federal system [66,67]. The GoN should recruit qualified health personnel with clear job descriptions [66], and invest in training and regular supervision, as was successfully done whilst implementing integrated nutrition and stimulation interventions in Kenya [56]. National, provincial and local governments should invest in in-service training for health workers, regular refresher training for FCHVs and supervisors' capacity to increase workforce capacity to deliver integrated interventions [45,58,65]. A structured monitoring system for evaluation of health service providers' performance and informing training and supervision would improve programme quality [45]. Financial and non-financial incentives to maximise the motivation of health service providers should be used [68].

While the devolution provides many opportunities for local governments to integrate stimulation intervention across sectors, there will be little progress until budgets and responsibilities are clearly defined. To expand stimulation interventions for children under three years into health and nutrition services as well as ECED centres, it is necessary to consider intersectoral and integrated policy framework to promote coordination and clarity between government bodies, particularly, health and education sectors, as well as across the levels of governance [33]. Nepal's new ECD strategy reflects a growing awareness of ECD as a priority and the importance of intersectoral collaboration [69]. However, ministries must be sensitised about ECD and commit to children's development [70].

## Strengths and limitations

This is the first study to examine Nepal's health system's readiness to integrate stimulation intervention into existing health and nutrition services. Strengths of this study include the triangulation of perspectives from a diverse group of participants, and multiple analysts independently coding data to create the coding framework. Limitations include that findings are susceptible to social desirability bias. For example, some participants may have said that integration is a good idea even though they foresaw challenges. We used a vignette to brainstorm participants' interpretations of stimulation. This may have indicated the importance of play to the participants and led them to share more positive views of stimulation than they would have otherwise. We collected data during the COVID-19 pandemic but did not investigate how the pandemic may have affected caregivers' participation in child health and nutrition services, or the capacity of health service providers to provide those services. The pandemic could have exacerbated the challenges in the community and the health system, although respondents did not specify this. Further research is needed to explore the consequences of COVID-19 on the possible integration of stimulation interventions into the health system to contextualise intervention strategies.

## Conclusions

The existing health system structure in Nepal provides opportunities for the integration of stimulation interventions into national health and nutrition programmes. The case for integration is strengthened by community members' largely positive perception about the role of stimulation in promoting ECD, existing health system structures to support integration, and a new ECD strategy to guide local governments. Efforts to develop a comprehensive and holistic approach to promoting ECD can be effective by taking a family-inclusive approach, strengthening the health system through continuous refresher training, supportive supervision, appropriate incentives, continuous monitoring and evaluation as well as building the capacity of the local governments. We encourage researchers, stakeholders, and policymakers to focus on overcoming the barriers while enhancing the facilitators to support the integration and implementation of a holistic approach to improve child development in rural Nepal.

## Supporting information

**S1 Table. Sampling framework.**
(PDF)

**S2 Table. Supporting qualitative data.**
(PDF)

**S1 File. Topic guides for semi-structured interviews and focus group discussions.**
(PDF)

**S2 File. Vignette: Children learn through play.**
(PDF)

**S3 File. Consolidated Criteria for Reporting Qualitative Research (COREQ) checklist: Exploring the feasibility of integrating health, nutrition and stimulation interventions for children under three years in Nepal's health system: A qualitative study.** Developed from: Tong A, Sainsbury P, Craig J. Consolidated criteria for reporting qualitative research (COREQ): A 32-item checklist for interviews and focus groups. International Journal for Quality in Health Care. 2007. Volume 19, Number 6: pp. 349–357.
(PDF)

## Acknowledgments

The authors thank the primary health care centre, health post, municipalities and ward offices from the study site in Dhanusha district for approving data collection and providing continuous support, without which this data collection would not have been feasible. We thank female community health volunteers from the study site for assisting us in communicating with the community members and identifying participants for the study. We wish to extend our gratitude to the community members, health service providers, governmental and non-governmental stakeholders and policymakers who volunteered their time, experiences, and perspectives during this project. We would also like to thank Shyam Sundar Yadav, Mahendra Paswan, Suresh Yadav and Ram Kumar Yadav from Unique Society Welfare Centre in Dhanusha district for supporting us with field transportation (via motorcycles) and communication with the local leaders; Deepa Thokar for assisting us in compiling information on current health and nutrition programmes in Dhanusha district; Ramesh Shrestha, for helping us with logistical arrangements for the data collection; and Bhim Prasad Shrestha, Ritesh Shrestha and Bishnu Bhandari from the Health Research and Development Forum in Kathmandu for their

guidance and support in preparing for the fieldwork. We would like to thank Dr Joanna Morrison for sharing her experiences and ideas about preparing topic guides for the interview.

## Author Contributions

**Conceptualization:** Sophiya Dulal, Audrey Prost.

**Data curation:** Sophiya Dulal.

**Formal analysis:** Sophiya Dulal, Naomi M. Saville, Dafna Merom, Audrey Prost.

**Funding acquisition:** Sophiya Dulal.

**Investigation:** Sophiya Dulal, Kalpana Giri.

**Methodology:** Sophiya Dulal, Audrey Prost.

**Project administration:** Sophiya Dulal, Kalpana Giri.

**Resources:** Sophiya Dulal.

**Software:** Sophiya Dulal.

**Supervision:** Naomi M. Saville, Audrey Prost.

**Validation:** Sophiya Dulal, Audrey Prost.

**Visualization:** Sophiya Dulal.

**Writing – original draft:** Sophiya Dulal.

**Writing – review & editing:** Sophiya Dulal, Naomi M. Saville, Dafna Merom, Audrey Prost.

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
