## [Decision Letter · Decision Letter 0]

21 Feb 2023

PGPH-D-22-01869

Exploring the feasibility of integrating health, nutrition and stimulation interventions for children under three years in Nepal’s health system: a qualitative study

Dear Dr. Dulal,

Thank you for submitting your manuscript to PLOS Global Public Health. After careful consideration, we feel that it has merit but does not fully meet PLOS Global Public Health’s publication criteria as it currently stands. Therefore, we invite you to submit a revised version of the manuscript that addresses the points raised during the review process.

We look forward to receiving your revised manuscript.

Kind regards,

Miquel Vall-llosera Camps

Staff Editor

Journal Requirements:

2. In the online submission form, you indicated that "The fully anonymised data will be shared on reasonable request with the corresponding author". All PLOS journals now require all data underlying the findings described in their manuscript to be freely available to other researchers, either 1. In a public repository, 2. Within the manuscript itself, or 3. Uploaded as supplementary information.

Reviewers' comments:

Reviewer's Responses to Questions

**Comments to the Author**

1. Does this manuscript meet PLOS Global Public Health’s publication criteria? Is the manuscript technically sound, and do the data support the conclusions? The manuscript must describe methodologically and ethically rigorous research with conclusions that are appropriately drawn based on the data presented.

Reviewer #1: Yes

Reviewer #2: Yes

2. Has the statistical analysis been performed appropriately and rigorously?

Reviewer #1: N/A

Reviewer #2: N/A

3. Have the authors made all data underlying the findings in their manuscript fully available (please refer to the Data Availability Statement at the start of the manuscript PDF file)?

Reviewer #1: No

Reviewer #2: Yes

4. Is the manuscript presented in an intelligible fashion and written in standard English?

Reviewer #1: Yes

Reviewer #2: Yes

5. Review Comments to the Author

Reviewer #1: Overall this an excellent paper. It tackles an important topic and is well designed, and based on evidence of a strong report, very well executed overall. It would make a valuable contribution to the literature and would be a good fit for PLOS GH.

My comments are thus very minor:

Sampling – can you say a little more about how the participants were identified. Were the local FCHVs know to you; did they know the families and what was chance they picked families who they thought might by pro-ECD? Likewise, were the policy-makers known to research team or was this the first interaction?

Please say a little more about SD and KG who (if I understood right) did the interviews. For example prior experience; background (see COREQ re ‘reflexivity of research team);

Did KG interpret all for SD or did SD also speak some Nepali? Was there discussion between SD and KG during interviews so as to steer next questions based on previous ones or did KG do all with SD just observing

Figure 1 is good – though resolution poor so I couldn’t see all details. Not essential, but consider making this more visually appealing with perhaps some further subheadings to summarise key points.

Discussion

Please say more about the potential role of ECD integrated in different TYPES of health programme. For instance, I can see a big opportunity when infants and children come for immunization visits or growth monitoring visits and may otherwise just be waiting around for quite a while before a short interaction with a HCW Likewise, there are long periods of inactivity during a CMAM nutrition clinic which could be well utilised by ECD activities and education. However, linking with IMCI may be more problematic and maybe not even desirable, at least in same way since by definition the children are unwell, clinical priorities should take 1st place and if sick, both children and parents are less likely to engage with ECD.

Reviewer #2: The article aims to understand the feasibility of integrating an ECD intervention for children below 3 years of age to the health system in Nepal. With the study the authors try to understand the barriers and enablers to fill an existing gap which is the delivery of ECD interventions for young children in Nepal. The manuscript has some important merits: A robust qualitative methodology that includes data collection from beneficiaries, providers, managers, and policy makers. Even though the findings have mainly a local interest, the authors raise an important aspect of the implementation barriers regarding cultural aspects, that are common to other countries. The manuscript is also very well written; however, it is very long. I suggest revising and trying to shorten, especially the results section.

There are minor issues that also should be addressed:

There is very limited information about the interviewers’ credentials, occupation, experience, and training. The authors didn’t include report the interviewer’s relationship with the participants if existent.

The sample size selection is briefly described, and the authors did not mention participants’ dropouts or refusals.

6. PLOS authors have the option to publish the peer review history of their article (what does this mean?). If published, this will include your full peer review and any attached files.

**Do you want your identity to be public for this peer review?** For information about this choice, including consent withdrawal, please see our Privacy Policy.

Reviewer #1: No

Reviewer #2: No

---

## [Decision Letter · Decision Letter 1]

11 Apr 2023

Exploring the feasibility of integrating health, nutrition and stimulation interventions for children under three years in Nepal’s health system: a qualitative study

PGPH-D-22-01869R1

Dear Ms Dulal,

We are pleased to inform you that your manuscript 'Exploring the feasibility of integrating health, nutrition and stimulation interventions for children under three years in Nepal’s health system: a qualitative study' has been provisionally accepted for publication in PLOS Global Public Health.

Before your manuscript can be formally accepted you will need to complete some formatting changes, which you will receive in a follow up email. A member of our team will be in touch with a set of requests. There is also one final comment from the reviewer (below) that you can incorporate at that time.

Best regards,

Julia Robinson

Executive Editor

Reviewer Comments (if any, and for reference):

Reviewer's Responses to Questions

**Comments to the Author**

1. If the authors have adequately addressed your comments raised in a previous round of review and you feel that this manuscript is now acceptable for publication, you may indicate that here to bypass the “Comments to the Author” section, enter your conflict of interest statement in the “Confidential to Editor” section, and submit your "Accept" recommendation.

Reviewer #2: All comments have been addressed

2. Does this manuscript meet PLOS Global Public Health’s publication criteria? Is the manuscript technically sound, and do the data support the conclusions? The manuscript must describe methodologically and ethically rigorous research with conclusions that are appropriately drawn based on the data presented.

Reviewer #2: Yes

3. Has the statistical analysis been performed appropriately and rigorously?

Reviewer #2: N/A

4. Have the authors made all data underlying the findings in their manuscript fully available (please refer to the Data Availability Statement at the start of the manuscript PDF file)?

Reviewer #2: Yes

5. Is the manuscript presented in an intelligible fashion and written in standard English?

Reviewer #2: Yes

6. Review Comments to the Author

Reviewer #2: The authors have addresses all the comments and have improved the quality of the article.

A minor correction is still needed in page 7, line 145 the authors have added information about the researchers knowledge of the iparticipants, using the researchers' names abbreviation, without identifying that the abbreviations refer to the researchers' names. The explanation only appears a few lines below in line 164, and should be presented where the abbreviations first appear.

7. PLOS authors have the option to publish the peer review history of their article (what does this mean?). If published, this will include your full peer review and any attached files.

**Do you want your identity to be public for this peer review?** For information about this choice, including consent withdrawal, please see our Privacy Policy.

Reviewer #2: **Yes: **Alexandra Brentani
